# Employer Actions in Office Settings and Women’s Perception of the Workplace as Supportive of Healthy Eating: A Cross-Sectional Pilot Study

**DOI:** 10.3390/nu16213766

**Published:** 2024-11-01

**Authors:** Aleksandra Hyży, Ilona Cieślak, Joanna Gotlib-Małkowska, Mariusz Panczyk, Alicja Kucharska, Mariusz Jaworski

**Affiliations:** 1Department of Education and Research in Health Sciences, Faculty of Health Science, Medical University of Warsaw, 00-581 Warsaw, Poland; aleksandra.hyzy@wum.edu.pl (A.H.); ilona.cieslak@wum.edu.pl (I.C.); joanna.gotlib@wum.edu.pl (J.G.-M.); mariusz.panczyk@wum.edu.pl (M.P.); 2Department of Human Nutrition, Faculty of Health Sciences, Medical University of Warsaw, 27 Erazma Ciołka Street, 01-445 Warsaw, Poland; alicja.kucharska@wum.edu.pl

**Keywords:** dietary habits, women’s health, office workers, health promotion, workplace environment, nutrition

## Abstract

Background/Objectives: This study aimed to evaluate how women working in office environments perceive their workplace as promoting healthy eating behaviors through employer-led actions. Methods: This cross-sectional study was conducted among 230 professionally active women employed in office settings in Poland. Data were collected using the Computer-Assisted Web Interview (CAWI) method. Participants were divided into two groups based on their perceived level of workplace support for healthy eating behaviors, as measured by the Workplace Healthy Eating Scale. Group 1 (n = 125; 54.3%; mean score = 15.69, SD = 3.76) and Group 2 (n = 105; 45.7%; mean score = 29.88, SD = 5.15) reflected low and high perceived support, respectively. Results: A linear regression model was employed to assess the association between the perceived level of support and specific workplace initiatives, including access to fresh fruits and vegetables, meal preparation facilities, cafeteria usage, lectures on nutrition, cooking workshops, and individual dietary consultations. For Group 1, access to fresh fruits and vegetables was the only factor significantly associated with a positive perception of the workplace as promoting healthy eating (*p* = 0.003), explaining 6.5% of the variance (adjusted R^2^ = 0.065). In Group 2, both access to fresh produce and participation in cooking workshops were significantly associated with positive workplace perceptions (*p* < 0.001), explaining 41% of the variance (adjusted R^2^ = 0.410). Conclusions: Access to fresh produce is a key determinant of employees’ perceptions of workplace support for healthy eating behaviors, with a notably greater impact observed when combined with additional activities such as cooking workshops. Employer-led initiatives focusing on practical dietary engagement appear to be effective in enhancing workplace perceptions of health promotion.

## 1. Introduction

In recent years, workplace health promotion has gained popularity, likely due to economic factors such as increased employee efficiency through reduced presenteeism (working while sick) and sick leave, as well as image-related factors such as employer branding or CSR/ESG actions [1]. Additionally, there is increased awareness regarding the importance of health promotion [2]. The WHO emphasizes that workplaces are ideal venues for health promotion, especially given the declining availability of healthcare services [1,3]. The concept of a health-promoting workplace is defined as a series of actions taken by employers aimed at strengthening and improving employee health [4]. One such initiative is the promotion of healthy eating behaviors. According to the current literature, a healthy lifestyle, including a nutritious diet, is positively linked to work engagement [5]. This concept originated in the United States, but in recent years, such initiatives have also been increasingly observed in Poland’s labor market. This is partly due to the presence of international companies and the growing interest in promoting health in the workplace, especially regarding healthy eating habits [6,7].

Creating a work environment that encourages healthy eating behaviors benefits employers, employees, and society at large. From the employer’s perspective, it leads to economic benefits like reduced presenteeism and absenteeism, as well as increased employee efficiency and productivity [8]. It also reduces employee turnover and boosts loyalty [2]. Promoting healthy eating can also serve as a marketing tool for employers, attracting talent and contributing to awards and recognition [2,7]. From a societal perspective, workplace health promotion, including nutritional education and awareness, can reduce the prevalence of chronic diseases, improve quality of life and quality of diet [9]; and even reduce health inequalities [3,10].

Offices are particularly common sites for introducing health-promoting interventions, likely due to organizational factors—most employees are present at the same time and experience fewer socioeconomic differences, such as education level and living conditions [6]. Moreover, demography plays a role, with some offices having a female-dominated workforce. Women, as they often have greater health-related competence, may have a stronger influence on whether a workplace is perceived as promoting healthy eating [11].

However, the literature on how employees, especially women, perceive workplace initiatives promoting healthy eating behaviors is limited. Most available research discusses various employer-led health programs, including those focused on both physical and mental health. Common activities include health-related lectures, workshops, and consultations with specialists. For nutrition, measures such as subsidies for healthy meals, access to fresh fruits and vegetables, workplace canteens, and workplace choice architecture modification for healthy behaviors are also implemented [12].

The type and number of offered activities vary and depend on the employer’s size, wealth, and policy [2,7,13]. The effectiveness of these initiatives is usually measured through changes in employee behavior or biochemical markers (e.g., glucose levels or lipid profiles) [14,15], but feedback from the participants themselves is rarely considered. Simply planning and implementing initiatives may not be enough if they do not meet the needs of the employees. Depending on the specific characteristics of a company, its demographic structure, industry, or location of operation, different resources or opportunities may be needed for employees. Additionally, the proper selection of activities may be crucial in terms of how the workplace is subjectively perceived as promoting healthy eating behaviors. Therefore, it is important to understand the needs and gaps among employees to ensure that the programs being developed are successful both economically and health-wise.

Our study considers variations in employees’ perceptions of employer support in the context of promoting healthy eating habits, aiming to enable a more precise understanding of the role of subjective assessment of this support in viewing the workplace as a setting that encourages healthy nutrition. The decision to divide participants into two groups, differing in their level of perceived support, is grounded in several significant premises.

First, there is evidence suggesting that subjective perceptions of support can substantially influence the level of engagement in health-related activities and, by extension, the perception of the workplace as health-promoting. However, it remains unclear whether this observation is also applicable to the promotion of healthy eating behaviors within the work environment. Scientific reports suggest that the workplace environment can influence employees to adopt healthier eating habits, for example, through observing others and the phenomenon of modeling [16], as well as implementing health-focused policies [17]. Second, differentiating based on perceived support may reveal important differences in employees’ expectations and needs, which could impact satisfaction with the employer’s initiatives [18]. Understanding these differences is essential to developing more individualized programs tailored to the needs of diverse groups within the organization. Third, categorizing by support levels offers the opportunity to examine how the same activities promoting healthy eating behaviors are perceived in two groups with different subjective evaluations of support.

This segmentation allows our study to more thoroughly explore potential factors affecting the effectiveness of implemented healthy eating programs, facilitating better alignment of strategies with employee needs and contributing to both health and economic benefits within the workplace.

This study aimed to evaluate how women working in office environments perceive their workplace as promoting healthy eating behaviors through employer-led actions. Two research questions were formulated:What types of initiatives promoting healthy eating behaviors are undertaken by employers?What factors influence women’s perception of their workplace as promoting healthy eating behaviors?

## 2. Materials and Methods

### 2.1. Design and Setting

A cross-sectional pilot study was conducted on 230 professionally active women working in offices. The inclusion criteria were as follows: (1) active employment for at least six months, (2) office work, (3) age over 18 years, (4) consent to participate, (5) female gender. Participants who did not meet these criteria were excluded. Finally, the participants were selected by the principal investigator (AH).

Data were collected using the CAWI (Computer-Assisted Web Interview) method, in which respondents filled out an electronic questionnaire. Each participant could complete the survey only once, and the data were collected anonymously. Participants came from various industries, including healthcare, social services, administration, and education.

### 2.2. Ethics Committee Approval

Participants were informed about the purpose of this study, the anonymity of their responses, and the voluntary nature of their participation. Each participant could withdraw consent at any stage without providing a reason. Ethical approval was obtained from the Ethics Committee of the Medical University of Warsaw (approval no. AKBE/291/2023).

### 2.3. Group Selection and Division

In designing our study, we accounted for limitations in access to complete statistical data on the employment of women engaged in office work in Poland. According to data from the Central Statistical Office, approximately 5.1 million people in Poland are employed in office-based roles, with women representing about 60% of this group, equating to approximately 3.6 million economically active women. Among this population, the majority are women performing office-related work. Based on the size of this sample and the estimated total population of economically active women in office employment, the calculated margin of error for our study was 6%, with a 95% confidence interval and an assumed proportion of 0.5.

Women working in office environments were divided into two groups based on their perception of the workplace as promoting healthy eating behaviors. The division was based on scores obtained on the Workplace Healthy Eating Scale, developed by A. Hyży (see Appendix A). The average score served as the dividing point. Respondents with scores below or equal to the average were classified into Group 1, while those with scores above the average were placed in Group 2.

As a result, Group 1 consisted of 125 respondents (54.3%) with an average score of M = 15.69 (SD = 3.76), while Group 2 included 105 respondents (45.7%) with an average score of M = 29.88 (SD = 5.15). The differences between the groups were statistically significant (Z = −13.067; *p* < 0.001). Women in Group 2 perceived their workplace more positively as promoting healthy eating behaviors compared to those in Group 1.

### 2.4. Research Tools

The Workplace Healthy Eating Scale, developed by A. Hyży, was used to assess how the workplace is perceived in terms of promoting healthy eating. It consists of 10 statements to which respondents respond on a 5-point Likert scale (1 = strongly disagree, 5 = strongly agree). An example item is “My employer supports me in making healthy dietary choices”. Scores range from 10 to 50 points, with higher scores indicating more positive perceptions of the workplace as promoting healthy eating behaviors (see Appendix A).

The scale demonstrated high reliability, with a Cronbach’s alpha of 0.902. Factor analysis showed that the scale is unidimensional and explains 53.758% of the variance. The KMO value was 0.904, indicating adequate sample size. Bartlett’s test of sphericity (χ^2^(105) = 1157.36, *p* < 0.001) confirmed significant interdependencies between variables.

Additionally, this study included an original questionnaire on employer initiatives aimed at promoting healthy eating behaviors. These actions were assessed based on several factors:Access to fresh fruits and vegetables at work (response options: never, very rarely, rarely, sometimes, often, and very often, where the answers were divided into two groups—‘never’ was counted as ‘no’, while all other answers were counted as ‘yes’);Availability of a space where meals can be prepared or heated (e.g., a social room or kitchen) (response options: never, very rarely, rarely, sometimes, often, and very often, where the answers were divided into two groups—‘never’ was counted as ‘no’, while all other answers were counted as ‘yes’);Use of a workplace canteen (response options: never, very rarely, rarely, sometimes, often, and very often, where the answers were divided into two groups—‘never’ was counted as ‘no’, while all other answers were counted as ‘yes’);Participation in nutrition lectures (response options: never, very rarely, rarely, sometimes, often, and very often, where the answers were divided into two groups—‘never’ was counted as ‘no’, while all other answers were counted as ‘yes’);Culinary workshops (response options: never, very rarely, rarely, sometimes, often, and very often, where the answers were divided into two groups—‘never’ was counted as ‘no’, while all other answers were counted as ‘yes’);Individual diet consultations (response options: never, very rarely, rarely, sometimes, often, and very often, where the answers were divided into two groups—‘never’ was counted as ‘no’, while all other answers were counted as ‘yes’).

Additionally, the respondents were characterized in terms of age; the number of days spent working on-site (from (1) none to (5) every working day); the number of hours spent on-site (from (1) less than 4, or between 4 and 8 (2), to (3) more than 8); type of employment contract ((1) employment contract, or (2) civil law contract (mandate, specific-task) to (3) B2B contract); meals consumed at work (breakfast (1), second breakfast (2), lunch (3), other (4)); self-preparation of meals for work (never (1), rarely (2), sometimes (3), often (4), very often or always (5)); using the employee canteen (never (1), rarely (2), sometimes (3), often (4), very often or always (5)); and access to a workplace kitchen (yes (1), no (2), I don’t know (3)).

### 2.5. Statistical Analysis

The chi-square test for qualitative variables or the Mann–Whitney U test for quantitative variables was used to assess differences between groups.

Linear regression analysis was used to evaluate the influence of the independent variable on the dependent variable, in which the dependent variable was office workers’ perception of the workplace as promoting healthy eating behaviors, while the independent variables were the following actions related to promoting a healthy diet in the workplace: access to fresh vegetables and fruits at the workplace, availability of a space to prepare or heat meals (e.g., break room, employee kitchen), use of an employee cafeteria, a lecture on food and nutrition, cooking workshops, and individual dietary consultations. The regression analysis was conducted while preserving the original coding of the independent variables (e.g., response options 1–6: never, very rarely, rarely, sometimes, often, very often), consistent with the response categories used in the research instrument. A *p*-value of less than 0.05 was considered statistically significant. Statistical analyses were conducted using the IBM SPSS statistical package (Version 28.0.1.0; IBM Corporation, SPSS Inc., Chicago, IL, USA).

## 3. Results

### 3.1. Respondents’ Characteristics

The average age of the women participating in this study was 29 ± 9.55 years (min = 20 years; max = 60 years). The analyzed groups did not differ in terms of age (Group 1: 29 ± 10.2 years vs. Group 2: 30 ± 8.8 years; Z = −0.942; *p* > 0.05).

In describing the group of respondents, it can be said that most were women employed under a work contract, working 4 to 8 h per day, 5 days a week. The average income varied greatly (Table 1). The study participants did not differ in terms of type of contract (Chi^2^ = 1.255; *p* > 0.05), number of workdays (Z = −0.936; *p* > 0.05), number of daily working hours (Chi^2^ = 2.091; *p* > 0.05), or average income (Chi^2^ = 4.472; *p* = 0.346).

### 3.2. Eating Behaviors at the Workplace

On average, the study participants consumed two meals at work (n = 125; 54.8%). The participant group included women who reported eating only one meal at work (n = 51; 22.2%) and those who reported eating four meals (n = 4; 1.7%). The groups did not differ in terms of the reported number of meals eaten at work (Z = −1.327; *p* = 0.185) (Table 2).

The participant group was very diverse regarding how often they prepared their own meals for work. The largest percentage of respondents stated that they prepared meals for work very often or always (n = 88; 38.3%), followed by often (n = 60; 26.1%), sometimes (n = 37; 16.1%), rarely (n = 29; 12.6%), and never (n = 16; 7.0%) (Table 2). The groups did not differ in terms of how often they prepared meals for work (Z = −0.940; *p* = 0.370).

The analyzed groups differed in terms of access to a place where meals could be prepared or heated (e.g., a break room or an employee kitchen) (Chi2 = 6.118; *p* = 0.047). Respondents from Group 1 (83.2%; n = 104) were less likely to report having access to such a place than those from Group 2 (91.4%; n = 96) (Table 2).

The groups also differed in terms of the frequency of using the employee canteen (Z = −4.344; *p* < 0.001). Respondents from Group 1 (20.8%; n = 26) were less likely to use the canteen than those from Group 2 (46.7%; n = 49) (Table 2).

### 3.3. Actions Taken by Employers in the Workplace to Promote Healthy Eating Behaviors

The study participants came from workplaces where, on average, one of the analyzed actions aimed at promoting healthy eating behaviors was implemented (M = 1.04; SD = 1.304). Almost half of the respondents indicated that no actions aimed at promoting healthy eating behaviors were implemented in their workplace (46.1%; n = 106). In the remaining workplaces, such actions were taken. The most common was one action aimed at promoting healthy eating behaviors at work (n = 65; 28.3%), followed by two actions (n = 23; 10%), three actions (n = 21; 9.1%), and four actions (n = 9; 3.9%).

The analyzed groups differed in terms of the reported number of actions taken by employers to promote healthy eating behaviors (Z = −5.121; *p* < 0.001; Group 1: M = 0.60 ± 0.852 vs. Group 2: M = 1.57 ± 1.537). Respondents from Group 2 reported more frequent access to such activities compared to Group 1 (Table 3).

### 3.4. Regression Analysis

Statistical analysis showed that only one factor had a statistically significant impact on the perception of the workplace as promoting healthy eating behaviors—access to fresh vegetables and fruits (Table 4). The other analyzed factors did not affect the dependent variable, i.e., the perception of the workplace as promoting healthy eating behaviors. However, it should be noted that the analyzed variable explained about 12% of the variance (adjusted R-squared = 0.119). The model was well fitted to the data and predicted the dependent variable better than the average: F(7;115) = 2.218; *p* = 0.038.

For Group 2, statistical analysis showed that two factors had a statistically significant impact on the perception of the workplace as promoting healthy eating behaviors—access to fresh vegetables and fruits and cooking workshops (Table 4). The other analyzed factors did not affect the dependent variable. However, it should be noted that the analyzed variables explained about 43% of the variance (adjusted R-squared = 0.435). The model was well fitted to the data and predicted the dependent variable better than the average: F(7;97) = 10.670; *p* < 0.001.

## 4. Discussion

### 4.1. General Insight into Promoting Healthy Eating in the Workplace

The promotion of healthy eating behaviors in the workplace was not common among the group of employees analyzed. Almost half of the respondents indicated that no such activities were undertaken in their workplace. When such activities were present, they usually involved just a single action, such as a lecture on food and nutrition or access to fruits and vegetables, rather than a series of coordinated actions forming a comprehensive program. This may be related to the challenges of implementing complex programs that promote healthy eating behaviors in the workplace. In the literature, it is emphasized that complex health programs are often less popular than isolated actions due to the difficulty of organizing them (needs analysis, planning interventions, higher costs, and difficult evaluation), lower engagement (large programs often target a group of employees rather than the entire organization), and results that are not immediately visible, as is the case with most lifestyle interventions [2,6]. Consequently, depending on the country, company, and sector, both individual interventions [19,20,21] and comprehensive workplace health promotion programs [14,22] can be found.

### 4.2. Impact of Fruit and Vegetable Access on Workplace Healthy Eating

In the groups analyzed, access to fruits and vegetables was the most common form of promoting healthy eating behaviors in the workplace and had a significant impact on the perception of the workplace as promoting healthy eating in both groups. According to an Antal report, around 24% of office workers in Poland currently have access to fresh fruits and vegetables at work [23]. These can be provided in various ways—once a week, several days a week, or daily. Additionally, this is not an expensive service, making it affordable even for smaller companies, thus making this benefit more accessible and egalitarian [13]. However, one might wonder if mere access to fruits and vegetables will significantly improve the eating habits of those benefiting from it. According to research by Geaney et al., changes in the composition of employee meals do increase the consumption of fruits and vegetables, but these changes are not substantial. Furthermore, the authors point to the low quality of data that would allow for a full assessment of the effectiveness of such measures [24]. Nevertheless, the availability of fresh fruits and vegetables was positively evaluated by respondents in the context of promoting healthy eating behaviors in the workplace. Daily access to fresh fruits and vegetables may foster healthy eating behaviors through social modeling, as described by Bandura [25]. Employees observing others taking advantage of free access to fruits and vegetables are more likely to do the same. In the long term, this is likely to have a positive impact on both their physical and mental health [26].

### 4.3. Impact of Culinary Workshop on Workplace Healthy Eating

In the second group, two factors were observed to influence the perception of the workplace as promoting healthy eating behaviors—access to fruits and vegetables (as already discussed) and access to cooking workshops provided by the employer. This is a relatively new form of promoting healthy eating behaviors at work, which can take the form of in-person or online sessions. Moreover, it has been noted that even single cooking workshops can have positive effects, such as increasing motivation to eat healthily and improving employees’ culinary skills [27,28]. Culinary skills are an important aspect of health literacy, often translating into better nutrition knowledge and improved meal planning and preparation [10].

Organizing cooking workshops requires more involvement and resources from employers than other activities, as it requires appropriate space, higher financial outlays (organization, sanitary requirements, ingredients, equipment, and a specialist to lead the workshop), and proper planning. As a result, such activities are usually more accessible to larger or wealthier companies, reducing their overall availability [13]. It is important to remember that cooking workshops should not only provide new knowledge but also serve as a source of motivation to adopt healthy eating behaviors. They involve simple culinary instruction and are not designed to turn participants into professional chefs. Therefore, it is important to carefully select the topics of these workshops (e.g., the types of dishes to be prepared). The literature emphasizes the creation of ‘teaching kitchens’, envisioned as ‘learning laboratories’, which can be used for such purposes, especially in medical institutions. In the future, companies could also take advantage of this concept [29].

The opportunity to participate in workshops at the workplace may encourage employees to try new foods, improve their culinary and health skills, and strengthen relationships between colleagues [30]. When planning culinary workshops, employers can draw from the growing field of culinary medicine. Culinary medicine is an evidence-based field of medicine that combines nutrition science and culinary arts to prepare healthy and tasty meals [31].

In the second group, cooking workshops had a significant impact on the perception of the workplace as promoting healthy eating behaviors, while this phenomenon was not observed in Group 1. It is worth considering what actions could be taken to encourage women in Group 1 to view cooking workshops as an important element of promoting healthy eating behaviors at work. The literature highlights employee engagement as a key element in the implementation of any activities in the workplace, including those promoting healthy eating. Conducting a thorough engagement survey would help assess employees’ needs and willingness to participate in personal and organizational development. The ability and willingness to participate in such workshops is also significant. If an employee (female) does not cook outside of work and considers cooking a waste of time, she is unlikely to take advantage of workshops on this topic at the workplace. This trend may be further reinforced by societal changes, where women are no longer as often the sole individuals responsible for cooking and caring for the family [10,11,30,32,33].

When considering additional aspects related to employee engagement and challenges in participating in culinary workshops, it is worth emphasizing that the success of such initiatives depends both on the care taken in organizing the event and on the level of employee engagement [34]. The literature emphasizes the importance of considering intrinsic motivation, such as individual interests in a healthy lifestyle [16,35,36]. To increase employee engagement in such initiatives, conducting a comprehensive analysis of staff needs and expectations is recommended [37]. The results of such a study could enable tailoring workshop content to participants’ preferences, potentially leading to higher acceptance and attendance rates. Flexibility in the workshop format can also be a key factor in enhancing accessibility; offering online options or scheduling sessions at different times of day allows for better alignment with employees’ schedules. For example, Asher et al. [38] showed that an online culinary nutrition course for health can be effective in improving eating behaviors.

### 4.4. Potential Factors Influencing Differences in Access to Workplace Nutrition-Related Activities

Differences in access to workplace nutrition-related activities could be observed between the groups. This may be due to the financial capacity of the employer (their ability to organize and implement the benefit) and their awareness (the need to organize and implement the benefit) [13]. The discrepancy in access to various activities is particularly evident between the public and private sectors, as well as between small and large companies [13]. Additionally, the type of position held may also be a factor—those working in higher positions within the organization are often the primary beneficiaries of activities promoting healthy eating, as they tend to have higher health literacy (from the outset) due to their higher education levels and material status [6]. This makes it even more important to ensure access to health-promoting activities for all employees, regardless of their position in the organization.

Another indicator of a workplace promoting healthy eating behaviors could be how employees prepare meals for work. Respondents in this study showed great variability in this regard. However, the regression analysis did not show an impact of access to workplace cafeterias or areas to prepare meals on employees’ perceptions of the workplace as promoting healthy eating behaviors. With rising employee expectations and employers’ desire to remain competitive, such spaces are becoming standard, and employees no longer view them as a benefit but almost as an employer’s obligation. Moreover, simply providing a space for meal preparation does not guarantee that employees will prepare healthy meals there. More important than the space itself are nutrition knowledge and the ability to compose and prepare healthy meals for work [8,9]. A lack of these skills may result in some employees ordering food to the office because they lack the time, knowledge, or skills to prepare meals themselves. The most natural explanation is the convenience of preparing and/or eating a meal on-site or the availability of facilities for ordering food to the office. This is supported by the Antal report, which further highlights the role of job position and income in preparing meals independently [16]. This approach is not only a cheaper solution but also offers more choice and opportunities for education and better nutritional decisions using appropriate tools [30,39].

### 4.5. Strengths and Limitations

This study was conducted on a diverse group of women performing office work. Additionally, it is one of the few studies that focuses on women, who represent the largest group among office workers but are often underrepresented in higher levels of organizations. This study included two groups that were similar to each other, including in terms of gross income, which is often highlighted in the literature as a factor influencing access to activities promoting healthy eating behaviors in the workplace and satisfaction with them.

Despite its strengths, this study is not without limitations—it did not take into account the sector of the company (public/private), the size of the company (number of employees), or the position of the respondents (intern/mid-level employee/manager, etc.), which may influence eating behaviors and the perception of the workplace as promoting health. However, obtaining this information in Poland, because of the huge taboo regarding money and income issues, is very difficult. Also, we would have had trouble with access to data to compare the results. Although such reports are being created, the access to them is limited. One needs to be responsible for HR policy in the company or be able to buy the report, which costs approx. PLN 10–12 thousand (approx. USD 3–4 thousand). Therefore, when we were preparing this study, we had to think of future problems we may encounter. We are working on solving this problem in future research.

### 4.6. Future Research Directions

One of the key directions for future research is to analyze the relationship between employer initiatives and women’s perception of the workplace as supportive of healthy eating, taking into account the sector (public vs. private) and company size. Existing evidence suggests that access to programs promoting healthy eating habits is strongly correlated with organizational characteristics, particularly its sector and financial resources. Therefore, future research should include an analysis of structural and financial differences between companies to develop more tailored and effective healthy eating programs, especially in the small and medium-sized enterprise sector. Unfortunately, due to the need to maintain full anonymity of the study participants, information on this factor was not collected.

Another essential area of research should be the evaluation of the long-term impact of programs supporting healthy eating on employee health. Current research mainly focuses on short-term effects, whereas future studies should examine long-term impacts, particularly on employees’ physical and mental health, the incidence of chronic diseases, productivity levels, and overall job satisfaction.

Regarding culinary workshops, an important research direction is to assess their impact on long-term changes in dietary habits and the development of participants’ culinary skills. The existing literature indicates short-term benefits from participating in culinary workshops; however, there is a lack of data on their lasting impact on culinary skills and daily food choices. Future studies could thus investigate to what extent regular participation in culinary workshops contributes to the long-term adoption of healthy eating habits.

## 5. Conclusions

Despite its growing popularity, workplace promotion of healthy eating behaviors remains underemphasized. Only half of workplaces implement such actions, and usually only single initiatives. The actions with the greatest impact on the perception of the workplace as promoting healthy eating behaviors include access to vegetables and fruits, as well as culinary workshops. Intensive educational efforts should be directed towards both employers and employees to increase the popularity of promoting healthy eating behaviors in the workplace. Additionally, employees should be effectively incentivized to participate in these initiatives, for example, through financial support, bonuses, or educational activities.

## Figures and Tables

**Table 1 nutrients-16-03766-t001:** The characteristics of the respondents based on the analyzed variables.

Variable		n	%
Number of days spent at the office	0	24	10.4
	1	19	8.3
	2	18	7.8
	3	22	9.6
	4	18	7.8
	5	129	56.1
Number of hours spent at the office	less than 4	6	2.6
	4 to 8	147	63.9
	more than 8	77	33.5
Type of contract	B2B contract	25	10.9
	civil law contract	39	17.0
	employment contract	166	72.2
Monthly salary (gross)	Less than PLN 3010	21	9.1
	PLN 3010–4999	51	22.2
	PLN 5000–6999	66	28.7
	PLN 7000–8999	34	14.8
	PLN 9000 and above	58	25.2

**Table 2 nutrients-16-03766-t002:** Eating behaviors at the workplace.

Eating Behaviors at the Workplace		Group 1 (n = 125)	Group 2 (n = 105)
		N (%)	N (%)
Number of meals consumed at work	1	26 (20.8%)	25 (23.8%)
	2	65 (52.0%)	61 (58.1%)
	3	32 (25.6%)	17 (16.2%)
	4	2 (1.6%)	2 (1.9%)
Self-preparation of meals for work	never	6 (4.8%)	10 (9.5%)
	rarely	19 (15.2%)	10 (9.5%)
	sometimes	19 (15.2%)	18 (17.1%)
	often	28 (22.4%)	32 (30.5%)
	very often or always	53 (42.4%)	35 (33.4%)
Access to a workplace kitchen	yes	104 (83.2%)	96 (91.4%)
	no	18 (14.4%)	5 (4.8%)
	I do not know	3 (2.4%)	4 (3.8%)
Use of employee canteen	never	99 (79.2%)	56 (53.4%)
	rarely	12 (9.6%)	12 (11.4%)
	sometimes	6 (4.8%)	16 (15.2%)
	often	5 (4.0%)	18 (17.1%)
	very often or always	3 (2.4%)	3 (2.9%)

**Table 3 nutrients-16-03766-t003:** Actions to promote healthy eating at the workplace.

Actions	Group 1 (n = 125)		Group 2 (n = 105)		Chi2	*p*
	Yes % (n)	No % (n)	Yes % (n)	No % (n)		
Lectures on food and nutrition	5.6% (7)	94.4% (118)	37.1% (39)	62.9% (66)	36.765	<0.001
Culinary workshop	4.0% (5)	96.0% (120)	14.3% (15)	85.7% (90)	11.635	0.020
Individual dietary consultations	3.2% (4)	96.8% (121)	14.3% (15)	85.7% (90)	10.898	0.012
Access to fresh fruits and vegetables	31.7% (41)	68.3% (84)	56.4% (61)	43.6% (44)	41.502	<0.001

**Table 4 nutrients-16-03766-t004:** Linear regression analysis.

	B	SE	Beta	t	*p*
Group 1—results on the Workplace Healthy Eating Scale were below and equal to average					
Constant	13.238	2.579		5.132	<0.001
Access to fresh fruits and vegetables at the workplace	0.726	0.240	0.279	3.027	0.003
Self-preparation of meals for work	−0.244	0.266	−0.082	−0.918	0.361
Use of the employee canteen	0.292	0.401	0.073	0.729	0.467
Access to a workplace kitchen	0.882	0.853	0.093	1.034	0.303
Individual dietary consultations	−0.750	1.556	−0.047	−0.482	0.631
Culinary workshops	0.219	0.869	0.025	0.253	0.801
Lectures on food and nutrition	0.404	0.464	0.082	0.871	0.386
Group 2—results on the Workplace Healthy Eating Scale were above average					
Constant	21.074	3.872		5.443	<0.001
Access to fresh fruits and vegetables at the workplace	0.384	0.191	0.16	2.006	0.048
Self-preparation of meals for work	0.222	0.330	0.055	0.673	0.503
Use of the employee canteen	−0.049	0.315	−0.012	−0.155	0.877
Access to a workplace kitchen	0.429	1.461	0.024	0.294	0.770
Individual dietary consultations	0.747	0.571	0.113	1.307	0.194
Culinary workshops	3.417	0.580	0.517	5.892	<0.001
Lectures on food and nutrition	0.258	0.337	0.072	0.766	0.445

## Data Availability

The Workplace Healthy Eating Scale (WHES) developed and used for this study is available on the *Nutrients* website.

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
