# Peer review of "Employer Actions in Office Settings and Women’s Perception of the Workplace as Supportive of Healthy Eating: A Cross-Sectional Pilot Study"

_nutrients, 2024, doi:10.3390/nu16213766_

Round 1
Reviewer 1 Report
Comments and Suggestions for Authors
1. the title “Analysis of Factors Influencing the Perception of Workplaces
Promoting Healthy Eating Behaviors by Women Working in Office Settings. Preliminary Research.” had two deficits. The one is the term “Promoting Healthy Eating Behaviors” is too blurry and not specific to the independent variables and dependent variable. The two is the style would be “...Settings: Preliminary Research”
2. The numerical data with decimal should use a dot “.” not a “,”. For example, 10,4 should be 10.4. The authors should amend them in all the manuscript.
3. Line 182, why “number of workdays (Z=-0.936; p>0.05)” was used? Did the authors use Mann--Whitney U test? If so, why others use chi-square test for similar data type? Likewise, line 190, for “number of meals eaten at work (Z=-1.327; p=0.185),” If the authors would like to compare two groups’ difference in such variables, please also present them by each group in Table style.
4. For 3.2.6 section, the Table 3 for the regression analysis should be listed for all predictors, not only significant ones. Besides, why did the “Nagelkerke R-squared” used? The logistic regression analysis was used?
5. For the regression analysis, why the authors take each the independent variable as “Yes” and “No” to execute the regression? Why did not use their original coding “1~6” to conduct the regression? 1=never, 2=very rarely, 3=... and the like.
6. From the research aims and literature as the authors wrote, the two groups setting, based on the perceived support, was not explained clearly. That is why did the authors decide to do such setting?
Author Response
Dear Reviewer,
We would like to express our sincere gratitude for your thorough analysis of our article and the invaluable comments, which have greatly contributed to enhancing our manuscript. All of the recommended revisions have been incorporated into the text and marked in blue to facilitate review. Detailed explanations addressing each of your comments are presented in the table as an attachment.
Please see the attachment

Reviewer 2 Report
Comments and Suggestions for Authors
This study offers valuable insights into an under-explored topic: the perceptions of healthy eating initiatives among female office workers. It makes a unique contribution to workplace health promotion literature, particularly within the Polish context. With the growing emphasis on employer-led health initiatives, this research has practical implications for improving employee well-being. However, certain issues require attention prior to publication:
1. The exclusion of key variables is a significant limitation of the article. As noted in the discussion, omitting factors such as company size, industry sector, and job position reduces the comprehensiveness of the analysis.
2. The discussion surrounding interventions like cooking workshops could be enhanced by delving further into employee engagement and identifying potential barriers to participation.
3. While the paper is generally well-organized, the use of additional subheadings and the division of longer paragraphs would improve clarity and readability in the discussion section.
4. Line 16: It is better to correct this sentence “Data were collected using the Computer-Assisted Web Interview method.” to “Data were collected using the Computer-Assisted Web Interview (CAWI) method”. Abbreviation is introduced in Materials and Methods, so it's better to introduce it here for consistency.
5. Line 37: Replace “like” with “such as”. “Such as” is more formal than “like” in academic writing.
6. Line 127: Replace “good” with “high”. The term “high” is more commonly used when referring to reliability scores.
7. Lines 352-353: It is better to correct this sentence “The promotion of healthy eating behaviors in the workplace, despite its growing popularity, still receives too little attention” to “Despite its growing popularity, workplace promotion of healthy eating behaviors remains underemphasized”.
8. Line 358: Using “incentivized” is more formal and precise in this context than “encouraged.”
Author Response
Dear Reviewer,
We would like to express our sincere gratitude for your thorough analysis of our article and the invaluable comments, which have greatly contributed to enhancing our manuscript. All of the recommended revisions have been incorporated into the text and marked in yellow to facilitate review. Detailed explanations addressing each of your comments are presented in the table as an attachment. Please see the attachment

Reviewer 3 Report
Comments and Suggestions for Authors
1. Title and abstract:
- In the title the design of the study should be mentioned clearly, not Preliminary Research, but better pilot study or cross sectional pilot study etc..
- The structure of the abstract is subsections, it should appear in one paragraph
- The keywords should not appear in title, should be different in order that the paper can be captured easily
2. Introduction
- The subsection 1.1 should be removed and to be figured within the main body of the introduction, as the latter should be concluded by a potential hypothesis of the article.
3. Methods
- A power analysis should be included to justify the sample size
4. Results
- The presentation of results may benefit from a better presentation, avoiding the very short subsections
5. Discussion
- The discussion section may benefit of adding also the new directions needed for future research on the topic.
6. References
- On a general scale the article is poorly referenced, please kindly enrich especially in the introduction and discussion sections
- Reference 16 should be or removed or better described
Author Response
Dear Reviewer,
We would like to express our sincere gratitude for your thorough analysis of our article and the invaluable comments, which have greatly contributed to enhancing our manuscript. All of the recommended revisions have been incorporated into the text and marked in green to facilitate review. Detailed explanations addressing each of your comments are presented in the table as an attachment. Please see the attachment.

Round 2
Reviewer 1 Report
Comments and Suggestions for Authors
OK